# Evaluating Targeted Therapies in Ovarian Cancer Metabolism: Novel Role for PCSK9 and Second Generation mTOR Inhibitors

**DOI:** 10.3390/cancers13153727

**Published:** 2021-07-24

**Authors:** Dafne Jacome Sanz, Juuli Raivola, Hanna Karvonen, Mariliina Arjama, Harlan Barker, Astrid Murumägi, Daniela Ungureanu

**Affiliations:** 1Laboratory of Immunoregulation, Faculty of Medicine and Health Technology, Tampere University, FI-33014 Tampere, Finland; dafne.jacomesanz@tuni.fi; 2Applied Tumor Genomics, Faculty of Medicine, University of Helsinki, FI-00014 Helsinki, Finland; juuli.raivola@helsinki.fi; 3Cancer Signaling, Faculty of Medicine and Health Technology, Tampere University, FI-33014 Tampere, Finland; hanna.karvonen@tuni.fi; 4Institute for Molecular Medicine, FIMM, University of Helsinki, FI-00014 Helsinki, Finland; mariliina.arjama@helsinki.fi (M.A.); astrid.murumagi@helsinki.fi (A.M.); 5Clinical Medicine 5, Faculty of Medicine and Health Technology, Tampere University, FI-33014 Tampere, Finland; harlan.barker@tuni.fi; 6Fimlab Ltd., Tampere University Hospital, FI-33520 Tampere, Finland

**Keywords:** omentum, ovarian cancers, PCSK9, drug testing, mTOR, metabolism, rapalogs

## Abstract

**Simple Summary:**

Ovarian cancer (OC) is known for its poor prognosis, due to the absence of reliable biomarkers and its late diagnosis, since the early-stage disease is almost asymptomatic. Lipid metabolism plays an important role in OC progression due to the development of omental metastasis in the abdominal cavity. The aim of our study was to assess the therapeutic role of various enzymes involved in lipid metabolism regulation or synthesis, in different subtypes of OC represented by cell lines as well as patient-derived cancer cell cultures (PDCs). We show that proprotein convertase subtilisin/kexin type 9 (PCSK9), a cholesterol-regulating enzyme, plays a pro-survival role in OC and targeting its expression impairs cancer cell growth. We also tested a small library of metabolic and mTOR-targeting drugs to identify drug vulnerabilities specific to various subtypes of OC. Our results show that in OC cell lines and PDCs the second generation of mTOR inhibitors such as AZD8055, vistusertib, dactolisib and sapanisertib, have higher cytotoxic activity compared to the first generation mTOR inhibitors such as rapalogs. These results suggest that, in the era of precision medicine, it is possible to target the metabolic pathway in OC and identify subtype-specific drug vulnerabilities that could be advanced to the clinic.

**Abstract:**

Background: Dysregulated lipid metabolism is emerging as a hallmark in several malignancies, including ovarian cancer (OC). Specifically, metastatic OC is highly dependent on lipid-rich omentum. We aimed to investigate the therapeutic value of targeting lipid metabolism in OC. For this purpose, we studied the role of PCSK9, a cholesterol-regulating enzyme, in OC cell survival and its downstream signaling. We also investigated the cytotoxic efficacy of a small library of metabolic (*n* = 11) and mTOR (*n* = 10) inhibitors using OC cell lines (*n* = 8) and ex vivo patient-derived cell cultures (PDCs, *n* = 5) to identify clinically suitable drug vulnerabilities. Targeting PCSK9 expression with siRNA or PCSK9 specific inhibitor (PF-06446846) impaired OC cell survival. In addition, overexpression of PCSK9 induced robust AKT phosphorylation along with increased expression of ERK1/2 and MEK1/2, suggesting a pro-survival role of PCSK9 in OC cells. Moreover, our drug testing revealed marked differences in cytotoxic responses to drugs targeting metabolic pathways of high-grade serous ovarian cancer (HGSOC) and low-grade serous ovarian cancer (LGSOC) PDCs. Our results show that targeting PCSK9 expression could impair OC cell survival, which warrants further investigation to address the dependency of this cancer on lipogenesis and omental metastasis. Moreover, the differences in metabolic gene expression and drug responses of OC PDCs indicate the existence of a metabolic heterogeneity within OC subtypes, which should be further explored for therapeutic improvements.

## 1. Introduction

Ovarian cancer (OC) continues to be the second most common gynecological cancer in the world, due to its late diagnosis and poor prognosis [1]. Molecular profiling has identified several histological subtypes with distinct biological and molecular properties; therefore, tailoring patient treatment is often complicated [2]. High-grade serous ovarian cancer (HGSOC) is the most common epithelial ovarian cancer (EOC) subtype, and is characterized by aberrant TP53, genomic instability, and homologous recombination (HR) defects in the DNA repair pathway in 50% of cases [3]. Other less common subtypes are low-grade serous ovarian cancer (LGSOC), mucinous, clear cell and endometrioid carcinoma, which are mostly driven by specific kinase mutations. KRAS and BRAF mutation are more common in LGSOC, KRAS mutation and HER2 amplification are found in mucinous EOC, and ARID1A mutations are found in clear cell EOC [4]. Platinum-based chemotherapy coupled with debulking surgery is the first-line of OC treatment; however, development of chemoresistance is highly prevalent among OC patients [3].

Altered cellular metabolism is one of the hallmarks of tumorigenesis, directly or indirectly connected to the accumulated genomic aberrations [5]. The metabolic fitness acquired by cancer cells will influence not just tumor evolution and plasticity, but also the tumor microenvironment, metastasis, and treatment outcome [6]. Cancer cells have enhanced metabolic efficiency and flexibility, allowing acquisition of necessary nutrients and the ability to use them effectively to sustain tumor growth, differentiation, and the development of metastasis [7]. This metabolic flexibility could be observed, for instance, during lipid scavenging in metastasis. Scavenging extracellular lipids, rather than increased lipogenesis, has emerged as an important adaptive mechanism bypassing the need to supply carbon to promote cancer cell growth [7]. This mechanism is commonly observed during omental metastasis, in which adipocytes provide fatty acids to tumor cells as a source of nutrients, particularly in OC [8,9].

A common site for metastatic disease arising from intraperitoneal tumors is the omental tissue which favors metastatic tumor growth [10]. Omentum, an organ primarily composed of adipocytes, provides a homing niche for abdominal tumor cell proliferation and metastasis [11]. Omental metastasis typically represents the largest intra-abdominal tumor for patients with advanced OC and 80% of all women with serous ovarian carcinoma present with omental metastases [8]. The adipocyte-rich omentum provides a proliferative advantage and transfers fatty acids to cancer cells as indicated by the high lipid content found in OC cells bordering the adipocytes in omental metastasis. Moreover, advanced OC patients are commonly presented with ascites, a reservoir of a complex mixture of soluble factors and cellular components, which provide a pro-inflammatory and tumor-promoting microenvironment for the tumor cells [12]. The ascites also facilitates omental metastasis, becoming a major source of morbidity and mortality for OC patients [13]. 

In this study, we investigated the role of PCSK9, the proprotein convertase subtilisin/kexin type-9 regulating cholesterol and lipid metabolism, in OC cell survival and its intracellular signaling, in cell lines (*n* = 8) and PDCs (*n* = 5). Our OC models are comprised of subtype-specific cell lines such as HGSOC and endometrioid, including paired cell-lines pre- and post- acquisition of cisplatin resistance, whereas PDCs were also chosen as representative models of HGSOC and LGSOC. We also analyzed the efficacy of a small library of anti-metabolic (*n* = 11) and first and second generation of mTOR inhibitors (*n* = 10) in order to identify metabolic vulnerabilities that could be therapeutically useful in OC treatment.

## 2. Materials and Methods

### 2.1. Cell Culture and Transfections

Human ovarian cancer (OC) cell lines OVCAR3 and OVCAR3cis were cultured in RPMI 1640 (Lonza, Basel, Switzerland) supplemented with 20% fetal bovine serum (Thermo Fisher Scientific, Waltham, MA, USA), 2 mM L-glutamine (Thermo Fisher Scientific) and penicillin/streptomycin (Thermo Fisher Scientific). The cisplatin resistant phenotype, OVCAR3cis, cells were cultured with 1 µM cisplatin (SelleckChem, Houston, TX, USA). The endometrial subtype OC cell lines A2780 and A2780cis, were cultured in RPMI 1640, supplemented with 10% fetal bovine serum, 2 mM L-glutamine and Primocin^TM^. The cisplatin resistant phenotype, A2780cis, cells were cultured with 1 µM cisplatin (Selleckchem). JHOS2 cells were cultured in DMEM/F-12 (Gibco^TM^, Thermo Fisher Scientific) supplemented with 10% fetal bovine serum, 1× MEM-NEAA (Thermo Fisher Scientific, Waltham, MA, USA) and Primocin^TM^ (InvivoGen, San Diego, CA, USA). Kuramochi and Ovsaho cell lines were cultured in RPMI 1640 supplemented with 10% fetal bovine serum, 2 mM L-glutamine and Primocin^TM^. COV362, HEK293T and HeLa cells were cultured in DMEM (Lonza), supplemented with 10% fetal bovine serum, 2mM L-glutamine and Primocin^TM^. All cell lines were kept and incubated at 37 °C and 5% CO2.

Small interference RNA (siRNA) transfections were performed with Transfection Reagent 1 or 4 (GE Dharmacon, Lafayette, CO, USA) cells and with small interference RNA (siRNA) ON-TARGETplus Non-targeting Control Pool (GE Dharmacon) acting as a control or ON-TARGET plus Human PCSK9 siRNA—SMART pool 10 nM (GE Dharmacon) were used to silence PCSK9 in ovarian cancer. The final concentration of siRNA used in transfection was 6 nM for OVCAR3 cells and 25 nM for HeLa cells

Human PCSK9/NARC1 plasmid (HG15813-CM) was transfected to HEK293T or JHOS2 cells with TurboFect™ Transfection Reagent (Thermo Fisher Scientific) and reduced serum media Opti-MEM I (Gibco^TM^, Thermo Fisher Scientific). pCINEO expression vector (Promega, Madison, WI, USA) was used as transfection control.

### 2.2. Cell Viability and Inhibitor Treatment

CellTiter-Glo (CTG) 2.0 (Promega) assay was used to measure cell viability, according to manufacturer’s instructions. Cells were plated to 96-well plates and untreated or treated with increasing concentrations of the PCSK9 inhibitor (PCSK9i) PF-06446846 hydrochloride (MedChemExpress, Monmouth Junction, USA) and viability was analyzed after 48 h. The luminescence was measured with Envision plate reader (PerkinElmer, Waltham, MA, USA).

### 2.3. Western Blot

Cells were lysed with 50 mM Tris-HCl (pH 7.5) containing 1 mM EDTA, 50 mM NaF, 150 mM NaCl, 1% Triton X-100, 10% glycerol plus 1% phosphatase and protease inhibitor cocktails (Bimake, Houston, TX, USA). Proteins were separated using SDS-PAGE and were transferred onto nitrocellulose membrane (Bio-Rad Laboratories, CA, USA). The following primary antibodies were used: anti-PCSK9 antibody (85813, Cell Signaling Technology, Danvers, MA, USA), pERK1/2 (T202/Y204) (Cell Signaling Technology, 9101), ERK1/2 (Cell Signaling Technology, 4696), pAKT (Ser473) (Cell Signaling Technology, 4060), AKT (Cell Signaling Technology, 2920), pMEK Ser217/221 (Cell Signaling Technology, 9121), MEK (Cell Signaling Technology, 4694) and anti β-tubulin (Santa Cruz Biotech sc-166729, Dallas, TX, USA). Specific LI-COR secondary antibodies IRDye^®^ 680RD Donkey anti-Goat IgG, IRDye^®^ 800CW Donkey anti-Mouse IgG, or IRDye^®^ 680RD Donkey anti-Rabbit IgG (LI-COR, Lincoln, NE, USA) were used before detection with Odyssey^®^ CLx Imaging System (LI-COR). Image analysis was performed using Image Studio ™ Lite software (LI-COR).

### 2.4. Drug Testing

Drug testing was performed for OC cell lines and PDCs with dose-dependent concentration of drugs for 72 h. Briefly, 1000–1500 cells were added to wells of 384-well plates with drugs pre-plated over a 10,000-fold concentration range (in five concentrations). After a 3-day incubation at 37 °C, cell viability was measured using the Cell Titer-Glo reagent (Promega) and plates were read with PHERAstar FS (BMG Labtech, Ortenberg, Germany). For each drug, dose–response curves were generated and DSS were calculated as previously described [14]. In short, DSS is a measure of drug–response based on the area under the dose–response curve that captures both the potency and the efficacy of the drug effect. It integrates complementary information extracted by half-maximal inhibitory concentration (IC50), slope and minimal and maximum asymptotes. Analysis of hierarchical clustering and similarity matrix of DSSs was performed using Morpheus software [15]. For drug combination, cells were pre-treated with 1 μM cisplatin before being added into drug-loaded 384 plates and cell survival was measured with Cell Titer-Glo after 72 h.

### 2.5. Gene Expression Analysis 

Total RNA was isolated from PDCs using total RNeasy kit (Qiagen, Tübinge, Germany). Quantity and quality of the RNA samples were assessed by Qubit (Thermo Fisher Scientific, USA) and Bioanalyzer (Agilent Technologies, Santa Clara, CA, USA). RNA with an RNA integrity number (RIN) > 8 was used to derive libraries and the sequencing was performed with Illumina HiSeq system (Illumina, San Diego, CA, USA) as previously described [16]. The expression of lipid metabolism and PCSK family genes was derived from RNA-seq analysis. Hierarchical clustering analysis was performed using log2 transformation of transcripts per kilobase million (TPM) values derived from RNA-Seq data from samples of five PDCs. Clustering was performed on genes involved in the lipid-metabolism pathway and PCSK family based on KEGG pathway definitions [17]. Clustering was performed with the cluster module of the SciPy library [18], using a Euclidean distance metric and Ward linkage variance. Visualization of clusters were performed with Matplotlib [19] and Seaborn [20]. 

## 3. Results

### 3.1. Cholesterol Modulating pCSK9 Proprotein Is Expressed in OC Cell Lines and Patient Samples 

Proprotein convertases (PCs) are serine proteases capable of specific proteolysis at the RXR/KR motif in a multitude of substrates such as growth factors and growth factor receptors and pro-angiogenic proteins [21]. This family of proteases consists of nine members: PC1/3, PC2, Furin, PC4, PC5/6, PACE4, PC7, SKI-1/S1P, and PCSK9 (NARC-1) [22]. The last member, PCSK9 (NARC-1) is expressed in adult liver, small intestine and ovary [23] and is known to regulate cholesterol metabolism through binding to low-density lipoprotein receptor (LDLR) and subsequent targeting of the receptor for lysosomal degradation [24,25]. Therefore, inhibition of PCSK9 plays an important role in hypercholesterolemia treatment [26]. Gain-of-function mutations in *PCSK9* increases LDLR degradation and, therefore, decreases the circulating LDL uptake and clearance [27]. Since lipid metabolism is highly deregulated in OC and acts as major mediator of omental metastasis, we wanted to assess whether PCSK9 expression could play a role in OC cell survival, both in cell lines and PDCs. For this purpose, we investigated PCSK9 expression in six OC cell lines (Table 1), comprised of HGSOC subtype cell lines (JHOS2, Ovsaho, Kuramochi, COV362, OVCAR3) as well as the endometrioid subtype A2780 cell line.

To date, there are not any known LGSOC representative cell lines. Moreover, for OVCAR3 and A2780 cell lines, we have included both parental and cisplatin-resistant clones, as platinum chemoresistance usually develops in OC patients and has therapeutic significance. Western blot analysis of OC cell line lysates identified variable expression of PCSK9 isoforms (uncleaved and cleaved) in comparison with cervical carcinoma HeLa cell line, which was previously shown to express PCSK9 [32] (Figure 1A,B and Appendix A). The highest PCSK9 expression among OC cell lines was identified in OVCAR3 cell lysates, whereas the cisplatin-resistant OVCAR3cis cell lysates showed lower PCSK9 levels. This difference, however, was not observed in the endometrioid subtype of A2780/A2780cis cell line lysates. Since cleaved PCSK9 is known to be secreted, we also analyzed cell supernatants and found enriched expression of cleaved PCSK9 in OVCAR3 and HeLa cell lines (Figure 1C and Appendix A).

Furthermore, we wanted to evaluate PCSK9 expression in OC tumor samples and for this, we used PDCs derived from HGSOC and LGSOC patient tumors. Personalized cancer therapy using ex vivo pharmacogenomic testing of patient-derived cancer cells has already shown great potential and PDCs are currently used as models for tumor molecular profiling as excellent ex vivo representatives of patients’ tumor [33]. We assessed PCSK9 expression levels in PDC cell lysates (Figure 1D,E and Appendix A) by Western blotting and observed that three out of five PDCs (PDC#2, PDC#4 and PDC#5) cell lysates were positive for uncleaved PCSK9, whereas LGSOC PDCs (PDC#4 and PDC#5) showed also detectable levels of cleaved PCSK9 (Figure 1D,E). Our results show for the first time that PCSK9 expression is detected in tumor-derived PDCs, indicating that modulation of cholesterol pathway could play a role in OC tumorigenesis. Moreover, we also analyzed the gene expression profile for both lipid-associated and PCSK family of proteins in PDCs. Hierarchical clustering of gene expression showed that the LGSOC PDCs (PDC#4 and PDC#5) clustered together while, likewise, expression values for HGSOC PDCs were more similar to each other (Figure 1F). Notably, we observed that compared to HGSOC, LGSOC PDCs had low expression of apolipoprotein family genes *APOA1**, APOA4, APOH, APOC3* as well as *PCSK2**, PCSK6* and *PCSK1*, suggesting a subtype specific metabolic gene expression profile of OC PDCs.

### 3.2. Targeting PCSK9 Expression Impairs OC Cell Viability 

Next, we wanted to assess whether targeting PCSK9 expression has any effect on OVCAR3 and HeLa cell viability. siRNA-mediated knockdown of PCSK9 expression resulted in a statistically significant reduction of cell survival (Figure 2A), and a robust downregulation of PCSK9 levels in cell lysates and the supernatant fraction of OVCAR3 cells (Figure 2B,C and Appendix A). Next, we investigated whether targeting PCSK9 with a specific inhibitor would similarly impair cancer cell survival. As shown in Figure 2D, PCSK9 inhibitor PF-06446846 had a clear effect on cell proliferation at 100µM concentration in HeLa, OVCAR3 and OVCAR3cis cells, whereas JHSO2 cells that do not express PCSK9 showed minimal sensitivity (85% cell survival), indicating that PCSK9 plays a survival role in these cancer cells. 

We also investigated whether overexpression of PCSK9 in JHOS2 cells that typically lack its expression will activate pro-survival signaling pathways such as AKT/ERK. Transient transfection of PCSK9 in JHOS2 cells resulted in increased phospho-AKT, phospho-ERK and phospho-MEK levels, as well as an increase in ERK and MEK basal expression (Figure 2E,F and Appendix A). This could indicate that the pro-survival role of PCSK9 in OC is, at least in part, mediated by activating intracellular AKT/MEK/ERK signaling. 

### 3.3. Anti-Metabolic Drugs Show OC Subtype-Specific Efficacies 

Metabolic plasticity allows cancer cells to thrive in environments defined by harsh conditions and targeting cancer cell metabolism could, therefore, be seen as a treatment strategy to improve cancer therapy [34]. Since we observed that targeting cholesterol-modulating PCSK9 enzyme with its specific PF-06446846 inhibitor could impair OC cancer cell survival, we decided to investigate the cytotoxic effect of other compounds involved in lipid-metabolism (Table 2) in OC cell lines and PDCs. Consecutively, we assessed the cytotoxicity of atorvastatin, a statin involved in cholesterol synthesis and previously shown to reduce tumor cell proliferation as neoadjuvant agent; erastin, a ferroptosis activator that causes excessive lipid peroxidation and cell death due to cysteine and glutathione depletion; pevonedistat, an inhibitor of NEDD8-activating enzyme (NAE) with potential antineoplastic activity; AZD3965 an inhibitor of monocarboxylate transporter 1 (MCT1) leading to accumulation of glycolytic intermediates; disulfiram-(CuCl_2_), an alcohol dehydrogenase inhibitor chelating copper ions selectively accumulated in cancer cells. We also focused on compounds that inhibit DNA or RNA synthesis by blocking the activity of various enzymes required for metabolic reactions, such as daporinad, AVN944, methotrexate, pemetrexed, TH588 and triapine (Table 2). Each compound was tested in 5 dilutions to generate a dose–response curve that was used to calculate a drug sensitivity score (DSS) [14], which is used for quantitative scoring of differential drug responses. A DSS ≥ 7 in at least one cell line was used as a threshold for drug efficacy evaluation. 

We observed variable responses among OC cell lines and PDCs to our diverse library of metabolic modifiers (Figure 3A,B and Appendix A). Notably, high DSS values were observed for daporinad, pevonedistat and AVN944 in both cell lines and PDCs, whereas OC cell lines were also more sensitive to methotrexate and pemetrexed. Interestingly, a high sensitivity to erastin was observed in PDC#3 derived from a HGSOC patient as well as in the Kuramochi cell line, and these results we also confirmed by hierarchical clustering of all DSS values for metabolic modifiers (Figure 3C). A similarity matrix and PCA-plot analysis of DSSs from cell lines and PDCs indicated that OVCAR3/OVCAR3cis and A2780/A2780cis clustered together, as well as LGSOC PDC#4 and PDC#5 (Figure 3D,E). On the other hand, HGSOC representative cell lines such as Kuramochi, Ovsaho, and JHOS2, as well as PDC#1, PDC#2 and PDC#3 (also HGSOC) formed their own cluster, suggesting that similar drug responses could be expected in subtype-specific OC models. 

### 3.4. mTOR Inhibitors Elicit Diverse Efficacies in OC Cell Lines and PDCs

The discovery and characterization of the mechanistic target of rapamycin (mTOR) pathway helped us to understand the molecular mechanisms that regulate eukaryotic cell growth, linking nutrient sensing to regulation of protein synthesis, metabolism, aging and disease development [53]. mTOR is a serine/threonine protein kinase that belongs to the PI3K-related kinase (PIKK) family and forms the catalytic subunit of two distinct protein complexes, known as mTOR Complex 1 (mTORC1) and 2 (mTORC2). mTORC1 consists of three core components: mTOR, Raptor (regulatory protein associated with mTOR), and mLST8 (mammalian lethal with Sec13 protein 8, also known as GßL) [54]. In addition, mTORC1 also contains two inhibitory subunits PRAS40 (proline-rich AKT substrate of 40 kDa) and DEPTOR (DEP domain containing mTOR interacting protein). mTORC2 contains mTOR, Rictor (rapamycin insensitive companion of mTOR) instead of Raptor, mLST8, and DEPTOR as well as the regulatory subunits mSin1 and Protor1/2 [54]. mTORC1 is a key regulator of the anabolic–catabolic balance in response to environmental conditions which acts by modulating protein, lipids and nucleotides synthesis to sustain cell growth [55]. On the other hand, mTORC2 controls cell proliferation and survival primarily by phosphorylating AKT, a key effector of insulin/PI3K signaling [56].

Targeting the mTOR pathway has been extensively explored in cancer therapy and the first mTOR inhibitors approved were a class of rapamycin derivatives known as “rapalogs”. Rapamycin, also known as sirolimus and one of the first generation mTOR inhibitors, exerts its effect by binding to FRBP-12 (12 kDa FK506-binding protein) to form a ternary complex with mTOR, leading to inactivation of mTORC1 [57]. Other rapalogs include everolimus, ridaforolimus, and temsirolimus, among others. We investigated the cytotoxic efficacies sirolimus, temsirolimus, everolimus, and ridaforolimus, as well as other tyrosine kinase inhibitors (second generation of rapalogs) targeting mTOR pathway, such as AZD8055, vistusertib, dactolisib, sapanisertib, CC-115 and CC-223 (Table 2) in OC cell lines and PDCs. Comparison of DSS values for all mTOR inhibitors in OC cell lines and PDCs showed high sensitivity for AZD8055, vistusertib, dactolisib and sapanisertib in all models except the COV362 cell line, which is known for its drug-resistant phenotype (Figure 4A,B and Appendix A). On the other hand, all rapalogs and CC-223 displayed moderate to low cytotoxic activity (Figure 4A,C). We also observed that LGSOC PDCs were in general more sensitive to mTOR inhibitors than HGSOC PDCs. Similarity matrix and principal component analysis (PCA) of cell lines and PDCs DSSs to mTOR inhibitors revealed that A2780/A2780cis and OVCAR3/OVCAR3cis clustered together, whereas LGSOC PDC#4 and PDC#5 clustered differently, indicating variable drug response profile within the same OC subtype (Figure 4D,E). In conclusion, the first generation of mTOR inhibitors, the rapalogs, did not show enhanced cytotoxic activity in OC cell lines and PDCs, whereas the more advanced mTOR inhibitors such as tyrosine kinase inhibitors were more efficient in killing OC cell lines and PDCs. 

## 4. Discussion

Many human diseases, including cancer, are the consequence of altered metabolic pathways. Cancer cells have distinctive metabolic phenotypes characterized by altered nutrient metabolism that supports the rapid manufacture of biomass required to sustain rapid cell proliferation [5]. OC is particularly known for its omental metastatic propensity and indeed, almost 80% of OC patients present metastasis in the omentum [8], suggesting a preferential site for metastatic OC tumors in this adipocyte-rich environment. Lipid-rich microenvironments such as omentum could provide cancer cells with a source of fatty acids for rapid tumor growth, and several studies have already provided ample evidence suggesting a strong role for fatty acid metabolism in tumorigenesis [58,59,60]. Furthermore, the expression and activity of various enzymes involved in the synthesis and catabolic pathways of fatty acids (phospholipids and cholesterol) are significantly dysregulated in cancers [61]. PCSK9, an enzyme involved in the regulation of LDL cholesterol homeostasis by targeting LDLR for degradation, plays an important role in the treatment of hypercholesterolemia [26]. Although dysregulated cholesterol levels are clearly associated with the pathogenesis of coronary artery disease, an association of cholesterol with cancer development has been already reported [62]. An early study indicated that women with high serum cholesterol levels have high risk of OC, suggesting a role for hypercholesterolemia in OC development [63]. 

In this study, we investigated expression levels and an anti-apoptotic role of PCSK9 in OC models including cell lines and PDCs from various subtypes of OC. We found variable PCSK9 expression in our OC cell lines, with OVCAR3 showing the highest level of PCSK9 (both, uncleaved and cleaved) compared to HeLa cells that are known to express PCSK9 (Figure 1A,B). Detectable levels of secreted PCSK9 were also observed in OVCAR3 and HeLa cell supernatants, suggesting that PCSK9 trafficking is intact in these cells (Figure 1C). Moreover, we also observed various levels of PCSK9 in PDCs, with three out of five PDCs showing detectable levels of uncleaved PCSK9, whereas in LGSOC PDC#4 and PDC#5 cell lysates we could also detect cleaved PCSK9 expression (Figure 1D and Figure 2E). More importantly, targeting PCSK9 expression with siRNA or the PF-06446846 inhibitor impaired cell proliferation (Figure 2A,D), suggesting that PCSK9 has a survival role in OC cells. In addition to its therapeutic role in regulating cholesterol levels, PCSK9 has been previously shown to be involved in cancer. For instance, PCSK9 enhances liver metastasis by maintaining high circulating cholesterol levels [64], whereas in gastric cancers high PCSK9 expression could promote metastasis development by activating the MAPK signaling pathway [65]. In line with this, we also show that overexpressing PCSK9 in OC cells could activate the AKT/MEK/ERK pathway (Figure 2E,F). Our results show for the first time that PCSK9 expression is detected in OC tumor-derived cancer cells or PDCs. Moreover, PCSK9 has a survival role for OC cells, indicating that this proprotein convertase could directly or indirectly modulate tumorigenesis and should be further investigated in cancer therapy. 

Our data also uncovered differences in lipid metabolism and PCSK family gene expression between LGSOC and HGSOC PDCs. Notably, lower levels of apolipoprotein family genes *APOA1*, *APOA4*, *APOH*, *APOC3* as well as *PCSK2*, *PCSK6*, *PCSK1* were observed in LGSOC PDCs compared to HGSOC PDCs (Figure 1E). Moreover, we also show detectable levels of cleaved PCSK9 in LGSOC PDCs (Figure 1D). Apolipoproteins are a family of multifunctional glycoproteins that bind and transport lipids in the circulatory system. Apolipoprotein A1 (ApoA1) is the major component of high-density lipoprotein (HDL) particles and is considered as atheroprotective [66] due to its role in fat and cholesterol clearance from blood. ApoA4, also known as Apoa-IV, is a lipid-binding protein and a component of HDL and chylomicrons with an active role in lipid absorption and metabolism, contributing to protection against diabetes and atherosclerosis [67]. Apolipoprotein H (ApoH), also known as β2-glycoprotein I (β2GPI) is a phospholipid-binding plasma protein that is involved in lipid metabolism, angiogenesis and atherogenesis, among others [68], whereas APOC3 or ApoC-III is an important regulator of triglyceride transport and dyslipidemia [69]. Taken together, our gene expression analyses show clear differences in lipid metabolism pathways in LGSOC versus HGSOC subtype. Low levels of apolipoproteins expression in LGSOC would indicate a lack of lipid binding and transport, which would result in impaired lipoproteins functions [70]. For instance, low ApoA1 levels have been observed before in LGSOC compared to HGSOC OC patient samples [71], which is in line with our current observation, and low levels of ApoA4 were also detected previously in the plasma levels of OC patients [72]. Our results are in line with recent studies on metabolic heterogeneity within OC subtypes, which identified clear differences of cellular metabolism between low-grade and high-grade disease, as well as between primary tumors and metastatic tissue [73,74].

Consequently, we sought to test a small library of anti-metabolic inhibitors as well as first and second generation mTOR inhibitors in various subtypes of OC models. Among our OC models, we included HGSOC and endometrioid cell lines, since LGSOC cell lines models have not been clearly identified thus far, as well as HGSOC and LGSOC PDCs that are closely related to the original tumor molecular profile. Our DSS analysis of metabolic inhibitors identified three drugs with overall good cytotoxic response in OC cell lines and PDCs: daporinad, pevonedistat and AVN944. Daporinad (APO866) is an inhibitor of nicotinamide phosphoribosyltransferase (NAMPT) specifically inhibiting the biosynthesis of NAD+ from niacinamide, which is essential for the cellular metabolism, protein modification and messenger synthesis [75]. Daporinad has been clinically investigated in advanced melanomas (NCT00432107, ClinicalTrials.gov) as well as chronic lymphocytic leukemia (CLL; NCT00435084, ClinicalTrials.gov) and cutaneous T-Cell Lymphoma (CTCL; NCT00431912). Interestingly, we observed high DSS values for daporinad with HGSOC PDCs but not LGSOC PDCs (Figure 3B), suggesting a subtype-specific response for this inhibitor in OC, and further studies should be conducted using more subtype OC models in order to substantiate our findings. AVN944 is an inhibitor of inosine monophosphate dehydrogenase (IMPDH) [37], an enzyme involved in the de novo synthesis of GTP required for DNA and RNA synthesis, resulting in cellular apoptosis due to lack of GTP [76]. AVN944 appears to have a selective effect on cancer cells that are more dependent on GTP production, whereas deprivation of GTP in normal cells results in a temporary slowing of cell growth only. AVN944 is being investigated for the treatment of patients with advanced hematologic malignancies (NCT00273936, ClinicalTrials.gov), since IMPDH is more expressed in hematological cancers, as well as for its anti-angiogenetic properties [77]. A recent study has documented that IMPDH2 is highly expressed in OC and may serve as a potential prognostic biomarker [78], suggesting that AVN944 could have good efficacy in OC treatment. Indeed, we observed a relatively good cytotoxic effect of AVN944 in all OC PDCs (DSS values ≥ 7, Figure 3B) as well as in all OC cell lines except COV362. Furthermore, we also observed enhanced cytotoxic activity of AVN944 in the presence of cisplatin treatment in PDC#2 and PDC#4 (Appendix A), suggesting an additive cytotoxic activity that should be clinically relevant. Taken together, these findings strongly support the use of AVN944 for the treatment of OC and should be further investigated in a clinical setting. Pevonedistat (MLN4924) is a second-generation inhibitor that targets NAE and subsequently blocks the neddylation-dependent activation of Cullin-RING E3 ligases (CRLs), leading to apoptotic cell death [79]. Pevonedistat is currently explored (alone or in combination) in several clinical trials for the treatment of acute myeloid leukemia (AML; NCT03268954, NCT04712942, NCT04266795, ClinicalTrials.gov) as well as of solid tumors including OC (NCT01862328, ClinicalTrials.gov) and non-small cell lung carcinoma (NSCLC; NCT03228186, ClinicalTrials.gov) and others. Previous studies in OC preclinical models have shown that pevonedistat treatment of mice bearing ovarian tumor xenografts derived from A2780/A2780cis cells significantly increased the efficacy of cisplatin against both cisplatin-sensitive and -resistant xenograft tumors [80]. Phase I clinical studies have shown better efficacy of pevonedistat in combination with standard-of-care chemotherapy agents such as platinum-based and taxane drugs, suggesting that pevonedistat might re-sensitize platinum-resistant patients to cisplatin treatment [81]. We observed a moderate sensitivity to pevonedistat treatment alone (DSS values ~10) in three out of five PDCs (both HGSOC and LGSOC), indicating a good starting point for combinatorial treatments including paclitaxel or cisplatin. In order to validate this hypothesis, we carried out a drug combination treatment of cisplatin and pevonedistat and noticed enhanced cytotoxic effect of this drug combination in PDC#2 and PDC#4 (Appendix A) compared to either drug treatments alone. Interestingly, we observed no difference in DSS values for paired cisplatin-resistant cell lines compared to their parental cell line (A2780/A2780cis and OVCAR3/OVCAR3cis), whereas HGSOC Kuramochi and Ovsaho cell lines were resistant to pevonedistat treatment (Figure 3A). Another interesting observation is the overall lack of cytotoxic effect of atorvastatin, a cholesterol lowering statin, in OC cell lines and PDCs (Figure 3A,B). Statins deplete intracellular cholesterol by reducing mevalonate synthesis, initiating transcriptional activation of sterol regulatory element-binding protein SREBP and, consequently, an increased membrane expression of LDLR, which leads to enhanced LDL cholesterol uptake from the bloodstream [82]. By contrast, PCSK9 controls LDLR turnover at the plasma membrane and inhibition of PCSK9 elicits the same cholesterol-lowering effect. The anti-tumor potential of statins has been highlighted in several studies showing reduced cancer risk, disease stage and recurrence in patients treated with statins [83]. However, not all cancer patients respond to statins treatment, suggesting that targeting the mevalonate pathway should be investigated in each cancer type as the complexity of this pathway regulation is cancer specific [84]. In our study, we observed that targeting PCSK9 expression impaired OC cell survival, suggesting that one way to efficiently kill PCSK9-expressing cancer cells is via its inhibition and this approach should be investigated in clinical trials in a similar way as statins. 

Among the mTOR targeted drugs, our screen showed that the first generation of rapalogs had overall low DSS values in both OC cell lines and PDCs, except for PDC#4 which displayed good sensitivity to all rapalogs (Figure 4B). Previous studies have shown a modest therapeutic effect of temsirolimus monotherapy in OC preclinical and clinical studies, indicating that combining rapalogs with other therapies would render better results [85]. Indeed, our drug combination studies of cisplatin and temsirolimus treatment in two PDC models (PDC#2 and PDC#4) showed better cytotoxic effect than either drug treatments alone. The second generation of mTOR inhibitors, which are kinase inhibitors, showed high activity in both OC cell lines and PDCs. All OC cell lines and PDCs achieved high DSS values (DSS ≥ 7) with AZD8055, vistusertib, dactolisib and sapanisertib, except COV362 which is known to have an overall drug-resistant profile. AZD8055 is a new ATP-competitive mTOR kinase inhibitor shown to have cytotoxic activity in OC cell lines alone [86,87] or in combination with trametinib [88]. Vistusertib is a dual mTORC1/2 kinase inhibitor that has been positively evaluated in a phase I clinical trial in combination with paclitaxel for HGSOC patients [89]. Dactolisib (NVP-BEZ235) is a dual mTOR/PI3K kinase inhibitor with previously demonstrated activity on OC preclinical models [49,90]. Sapanisertib (TAK 228), also a dual mTORC1/2 kinase inhibitor, has been evaluated in a phase Ib trial in combination with serabelisib (TAK 117) and paclitaxel in patients with advanced solid tumors, including OC [91], indicating good anti-tumor activity and tolerability. Therefore, investigation of sapanisertib as a single agent and in novel treatment combinations is warranted.

## 5. Conclusions

Our study is the first to evaluate the expression and anti-apoptotic role of PCSK9 in OC. By investigating the efficacy of various metabolic and mTOR inhibitors ex vivo using tumor representative OC PDCs, our results showed differences in drug cytotoxic responses in LGSOC compared to HGSOC PDCs, which is in line with previous studies showing that OC subtypes display clear metabolic phenotypes [73]. As such, more studies should be directed towards a better understanding of the metabolic profile between various OC subtypes, which could help in tailoring better therapeutic benefits.

## Figures and Tables

**Figure 1 cancers-13-03727-f001:**
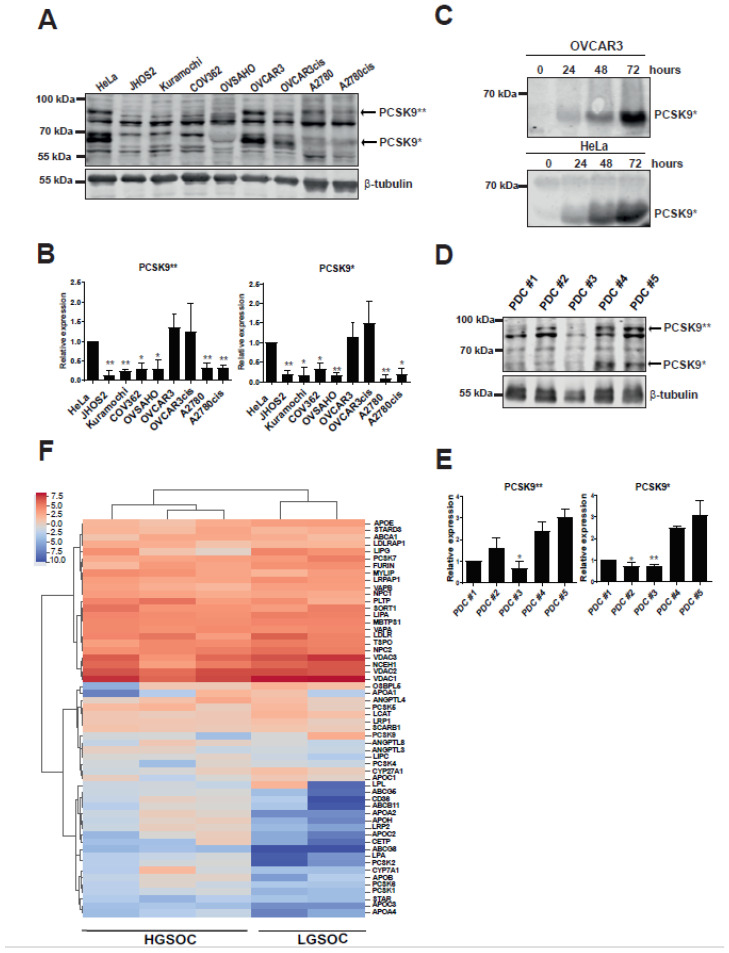
PCSK9 proprotein is expressed in OC. (**A**) PCSK9 protein expression in OC cell lines. Cervical carcinoma HeLa cell line served as positive control. Arrows indicate the molecular weight of cleaved (PCSK9*) and uncleaved (PCSK9**) isoforms. β-tubulin was used as loading control. (**B**) Quantification of PCSK9* and PCSK9** isoform levels from A. Protein levels were normalized to β-tubulin and then compared to PCSK9 isoform levels in HeLa cells (set to 1 for reference point, error bars indicate -/+ SEM of 3 independent experiments. Two-tailed student´s t-test was used to calculate statistical significance, which was indicated as * (*p* ≤ 0.05) or ** (*p* ≤ 0.01). (**C**) PCSK9* expression in supernatants of OVCAR3 and HeLa cells detected at various time points as indicated. (**D**) Western blot analysis of PCSK9* and PCSK9** isoform expression in lysates of OC PDCs. (**E**) Quantification of PCSK9* and PCSK9** isoform levels from (**D**). An arbitrary value of 1 was given to PCSK9 isoform levels in PDC#1. Error bars indicate −/+ SEM of 3 independent experiments. The statistical analysis was performed as in (**B**). (**F**) Hierarchical clustering of KEGG defined PCSK family and lipid-associated gene expression pathways of the low-grade serous ovarian carcinoma (LGSOC) and high-grade serous ovarian carcinoma (HGSOC) PDCs. Values are presented as log2 transformed transcripts per kilobase million (TPM) from RNA-seq from five PDCs. The uncropped Western Blot images can be found in Appendix A, and signal values are listed in Appendix A.

**Figure 2 cancers-13-03727-f002:**
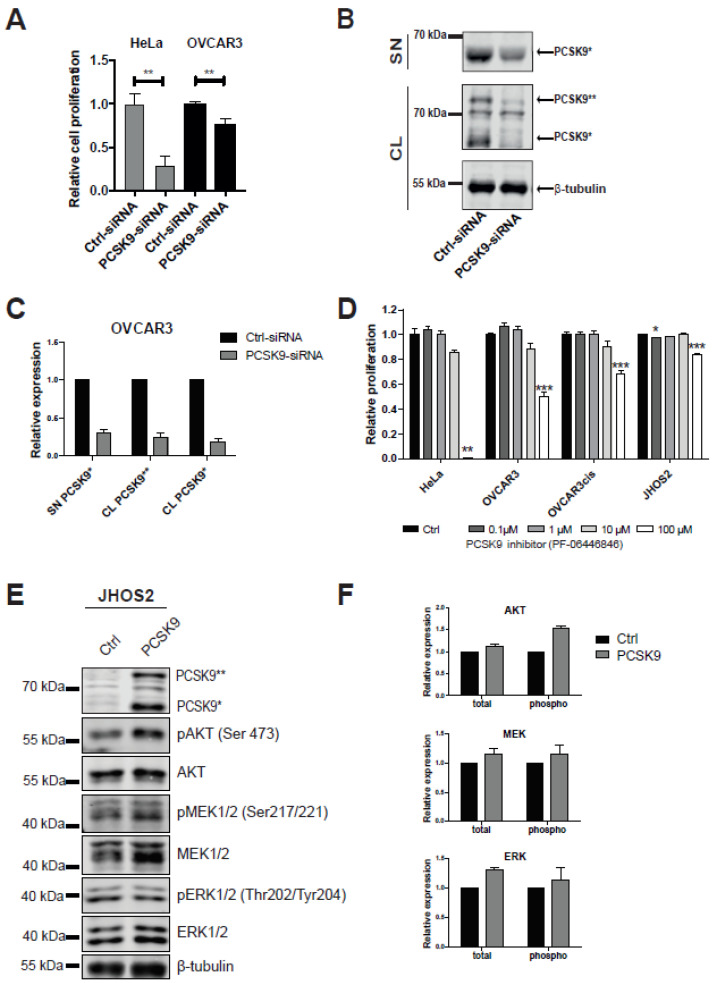
Targeting PCSK9 expression impairs OC cell viability. (**A**) Relative cell proliferation of OVCAR3 and HeLa cells treated with control or PCSK9-siRNA for 48 h. Statistical significance was tested using a two-tailed *t*-test (* indicates *p*-value <0.05 and *** indicates *p*-value <0.001). (**B**) Western blot analysis of PCSK9 isoform levels following 48 h siRNA treatment from OVCAR3 supernatant (SN) and cell lysate (CL). (**C**) Quantification of PCSK9** and PCSK9* isoform levels from (B). An arbitrary value of 1 was given to PCSK9 isoform levels in control siRNA treated cells. Error bars indicate −/+ SEM of 3 independent experiments. (**D**) Relative cell survival of HeLa, OVCAR3, OVCAR3cis (PCSK9-positive) and JHOS2 cells (PCSK9-negative) treated with increasing amounts of PCSK9 inhibitor PF-06446846 for 48 h. Statistical significance was calculated using two-tailed *t*-test (* indicates *p*-value < 0.05; ** indicates *p*-value < 0.01; *** indicates *p*-value < 0.001). (**E**) Western blot analysis of JHOS2 cell lysates non-transfected or transfected with PCSK9 for 48 h. (**F**) Quantification of phosphorylated and total AKT, ERK and MEK protein levels from (**E**). Arbitrary value of 1 was given to the indicated protein levels in control (non-transfected) sample. Error bars indicate -/+ SEM of 3 independent experiments. The uncropped Western Blot images can be found in Appendix A and signal values are listed in Appendix A.

**Figure 3 cancers-13-03727-f003:**
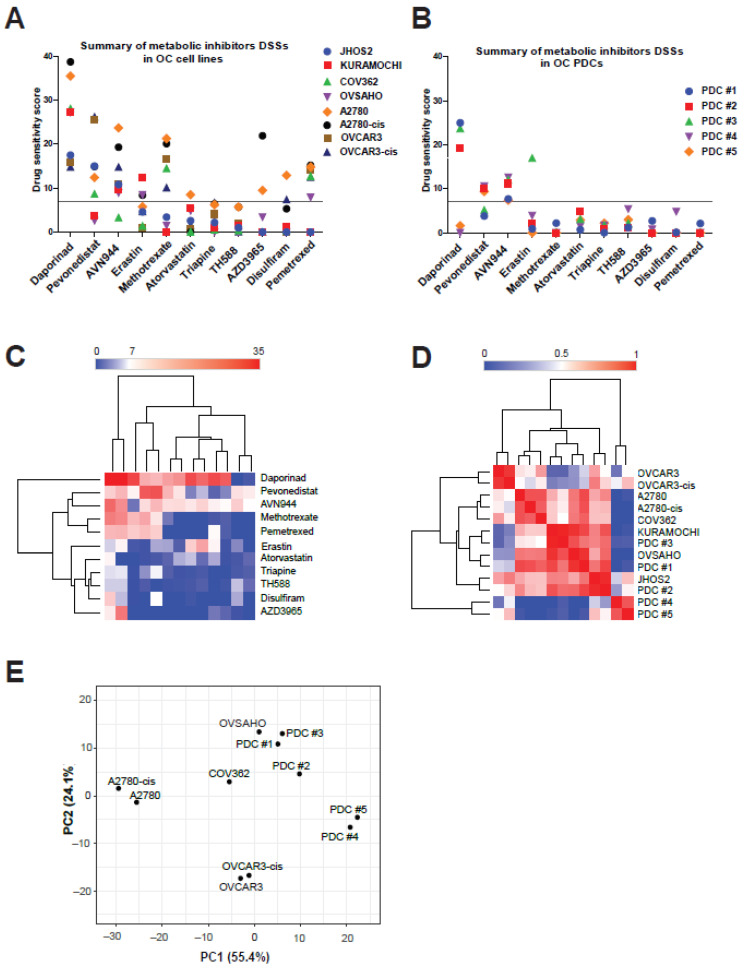
Evaluation of drug responses for anti-metabolic drugs (*n* = 11) in OC cell lines and PDCs. Comparison of DSS values for anti-metabolic drugs in OC cell lines (**A**) and PDCs (**B**). An arbitrary threshold of DSS 7 was chosen, as it indicates a moderate drug cytotoxicity. Analysis of drug responses based on hierarchical clustering (**C**), similarity matrix (**D**) and PCA plot (**E**) of combined DSS values from OC cell lines and PDCs (from patients #1 to #5).

**Figure 4 cancers-13-03727-f004:**
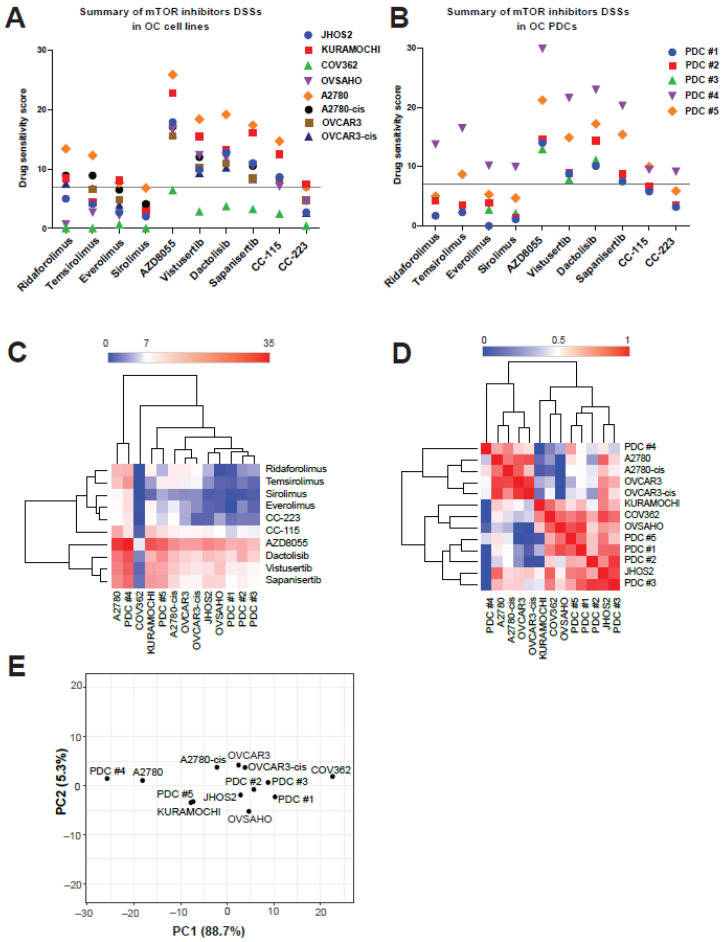
Evaluation of drug responses for mTOR inhibitors (*n* = 10) in OC cell lines and PDCs. Comparison of DSS values for mTOR inhibitors in OC cell lines (**A**) and PDCs (**B**). An arbitrary threshold of DSS 7 was chosen to define a moderate drug cytotoxicity. Analysis of drug responses based on hierarchical clustering (**C**), similarity matrix (**D**) and PCA plot (**E**) of combined DSS values from OC cell lines and PDCs (from patients #1 to #5).

**Table 1 cancers-13-03727-t001:** Summary of cell lines and patient-derived cell lines used in these studies. Abbreviations: mut: mutation, amp: amplification, fs: frameshift, del: deletion.

Name	Cell Type	Main Genomic Aberrations
HeLa	Endometrial carcinoma	
JHOS2	High grade ovarian serous adenocarcinoma.	*TP53* mut; BRCA1 mut [28,29]
Kuramochi	High grade ovarian serous adenocarcinoma	*TP53* mut; BRCA2 mut; MYC amp; KRAS amp
COV362	High grade ovarian serous adenocarcinoma	*TP53* mut; BRCA1 mut; MYC amp; RB1 del
Ovsaho	High grade ovarian serous adenocarcinoma	*TP53* mut; RB1 del; BRCA1 del [28,29]
OVCAR3	High grade ovarian serous adenocarcinoma	*TP53* mut [30]
OVCAR3cis	High grade ovarian serous adenocarcinoma, resistant to cisplatin	*TP53* mut [30,31]
A2780	Ovarian endometrioid adenocarcinoma.	PIK3CA mut; PTEN mut; BRAF mut; ARD1A mut [28,29]
A2780cis	Ovarian endometrioid adenocarcinoma, resistant to cisplatin	
PDC #1	Patient-derived cell culture (PDCs) from high grade serous OC tumor	*TP53* mut p.R175H; CCNE1 amp; PAX8 positive
PDC #2	Patient-derived cell culture (PDCs) from high grade serous OC tumor	*TP53* mut p. R283P; CCNE1 amp; PAX8 positive
PDC #3	Patient-derived cell culture (PDCs) from high grade serous OC tumor	*TP53* fs, MYC amp; KRAS amp; PAX8 positive
PDC #4	Patient-derived cell culture (PDCs) from low grade serous OC tumor	*TP53* WT; CDKN2A homozygous loss; PAX8 positive
PDC #5	Patient-derived cell culture (PDCs) from low grade serous OC tumor	*TP53* WT; CDKN2A homozygous loss; PAX8 positive

**Table 2 cancers-13-03727-t002:** Metabolic modifiers and mTOR inhibitors investigated in this study. Selected reference, biochemical class and main clinical trials are also listed. Clinical trial data is from DrugBank database [35].

Drug Name	Mechanism/Targets	Biochemical Class	Clinical Trials *(OC)*
Daporinad [36]	Targets nicotinamide phosphoribosyltransferase (NAMPT), an intermediate in the biosynthesis of nicotinamide adenine dinucleotide (NAD)	Metabolic modifier	Melanoma (Phase II), B-cell Chronic Lymphocytic Leukemia (Phase I/II), T-cell Lymphoma (Phase II)
Pevonedistat [36]	Inhibitor of Nedd8 activating enzyme (NAE)	Metabolic modifier	Myelodysplastic Syndrome (Phase I), Acute Myeloid Leukemia (Phase I/2), Advanced/Solid Tumors (Phase I), Multiple Myeloma (Phase I), Melanoma (Phase I)
AVN944 [37]	Inhibitor of inosine monosphosphate dehydrogenase (IMPDH), an enzyme involved in the de novo synthesis of GTP	Metabolic modifier	Hematological Malignancies (Phase I)
Erastin [38]	Activator of ferroptosis by acting as a voltage-dependent anion channel (VDAC) inhibitor; depletes cellular cysteine and glutathione, inducing excessive lipid peroxidation and cell death	Metabolic modifier	-
Methotrexate [39]	Inhibits dihydrofolate reductase enzyme, resulting in inhibition of purine nucleotide and thymidylate synthesis	Metabolic modifier	Ovarian cancer (Phase II), Acute Lymphoblastic Leukemia (Phase IV), Rheumatoid arthritis (Phase IV); Psoriasis (Phase IV), Breast Cancer (Phase II)
Atorvastatin [40]	Statin, inhibits hepatic hydroxymethyl-glutaryl coenzyme A (HMG-CoA) reductase involved in cholesterol synthesis	Metabolic modifier	Cardiovascular Disease (Phase IV), Cholesterol LDL (Phase IV), Type 2 Diabetes Mellitus (Phase IV), Alzheimer’s Disease (Phase III)
Triapine [41]	Inhibitor of ribonucleotide reductase (RNR); DNA synthesis inhibitor	Metabolic modifier	Ovarian Epithelial Cancer (Phase II), Leukemia/ Myelodysplastic Syndromes (Phase I/II) Lung Cancer (Phase II), Prostate Cancer (Phase II), Adenocarcinoma (Phase II)
TH588 [42]	Inhibitor of mut-T homolog-1 (MTH1, also known as NUDT1) that eliminates oxidized dNTP pools to prevent incorporation of damaged bases during DNA replication; impairs mitotic progression and mitotic DNA synthesis	Metabolic modifier	-
AZD3965 [43]	Monocarboxylate transporter 1 (MCT1) inhibitor; impairs lactate efflux leading to accumulation of glycolytic intermediates	Metabolic modifier	-
Disulfiram (+CuCl2) [44]	Alcohol dehydrogenase inhibitor chelating Cu+ selectively accumulated in cancer cells; generates reactive oxygen species (ROS) and inhibits proteasome activity	Metabolic modifier	Alcohol Dependence (Phase IV), Opioid Dependence (Phase II), Melanomas (Phase II), Glioblastoma (Phase II/III), HIV Infections (Phase I/II)
Pemetrexed [45]	Inhibitor of thymidylate synthase (TS) enzyme involved in DNA synthesis	Metabolic modifier	Ovarian cancer (Phase II), Lung cancer (Phase I), Breast cancer (Phase I), Prostate cancer (Phase II).
Ridaforolimus [46]	mTORC1 inhibitor, binds peptidyl-prolyl cis-trans isomerase FKBP12	Rapalog	Ovarian Cancer (Phase I), Metastatic sarcomas (Phase III), Breast cancer (Phase II), Prostate cancer (Phase II), Lymphoma/Myeloma (Phase II), Lung Cancer (Phase II)
Temsirolimus [46]	mTORC1 inhibitor, binds peptidyl-prolyl cis-trans isomerase FKBP12	Rapalog	Ovarian cancer (Phase II), Pancreatic Cancer (Phase II), Acute Myeloid Leukemia (Phase II), Glioblastoma (Phase II), Sarcomas (Phase II), Renal Cancers (Phase I), Breast cancer (Phase I/II)
Everolimus [46]	mTORC1 inhibitor, binds peptidyl-prolyl cis-trans isomerase FKBP12	Rapalog	Ovarian Cancer (Phase II), Breast cancer (Phase IV), Allograft rejection (Phase IV),
Sirolimus [47]	mTORC1 inhibitor, binds peptidyl-prolyl cis-trans isomerase FKBP12	Rapalog	Ovarian cancer (Phase II), Allograft rejection (Phase IV), Leukemia (Phase III), Hepatocellular Carcinoma (Phase III), Breast Cancer (Phase II)
AZD8055 [48]	mTOR inhibitor	Kinase inhibitor	Advanced Solid Malignancies (Phase I)
Vistusertib [48]	mTOR inhibitor, ATP-competitive	Kinase inhibitor	Breast Cancer (Phase I), Prostate Cancer (Phase I), Lung Cancer (Phase I)
Dactolisib [49]	mTOR/(PI3K) inhibitor	Kinase inhibitor	Breast Cancer (Phase I), Prostate Cancer (Phase I)
Sapanisertib [50]	mTOR inhibitor	Kinase inhibitor	Breast Cancer (Phase I), Prostate Cancer (Phase I), Lung cancer (Phase I)
CC-115 [51]	mTOR/DNA-dependent protein kinase (DNA-PK) inhibitor	Kinase inhibitor	-
CC-223 [52]	mTOR inhibitor	Kinase inhibitor	Hepatocellular Carcinoma (Phase I/II), B-Cell Lymphoma (Phase I/II), Glioblastoma (Phase I/II), Lung Cancer (Phase I/II)

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
