# Peer review of "Evaluating Targeted Therapies in Ovarian Cancer Metabolism: Novel Role for PCSK9 and Second Generation mTOR Inhibitors"

_cancers, 2021, doi:10.3390/cancers13153727_

Round 1

Reviewer 1 Report

The manuscript by Dafne Jacome Sanz titled Evaluating targeted therapies in ovarian cancer metabolism: novel role for PCSK9 and second generation of mTOR inhibitors. The authors showed that targeting proprotein convertase 22 subtilisin/kexin type 9 (PCSK9), a cholesterol-regulating enzyme, inhibits proliferation and survival in vitro and metastasis in vivo.  In addition, a small library of metabolic and mTOR targeted drugs were screened in a number of cell lines.  The second generation of mTOR inhibitors such as AZD8055, vistusertib, dactolisib and sapanisertib have higher cytotoxic activity compared to the first generation mTOR inhibitors. These results suggest that targeting metabolic pathways in ovarian cancer in the clinic should be explored.  The authors should validate the screening results, and test in combination. Mouse models should be used to validate these results and/or the combination of these treatments with stand of care drugs should be explored. The manuscript in its present state does not contain enough data and impact for publication as is. 

Author Response

Point by point answer to reviewers:

We would like to thank both reviewers for their insightful comments that strengthened our manuscript and made it more ready for publication. We have addressed all the comments/questions raised by the reviewers and the detailed answers is presented below in red. Also, the new edits made in the manuscript are marked by red for visibility.

Reviewer # 1:

The manuscript by Dafne Jacome Sanz titled Evaluating targeted therapies in ovarian cancer metabolism: novel role for PCSK9 and second generation of mTOR inhibitors. The authors showed that targeting proprotein convertase 22 subtilisin/kexin type 9 (PCSK9), a cholesterol-regulating enzyme, inhibits proliferation and survival in vitro and metastasis in vivo.  In addition, a small library of metabolic and mTOR targeted drugs were screened in a number of cell lines.  The second generation of mTOR inhibitors such as AZD8055, vistusertib, dactolisib and sapanisertib have higher cytotoxic activity compared to the first generation mTOR inhibitors. These results suggest that targeting metabolic pathways in ovarian cancer in the clinic should be explored.  The authors should validate the screening results, and test in combination. Mouse models should be used to validate these results and/or the combination of these treatments with stand of care drugs should be explored. The manuscript in its present state does not contain enough data and impact for publication as is.

Answer to Reviewer #1:

We thank the reviewer for his valuable comments and his suggestions on how to improve the manuscript.

We provide now more data using combinatorial drug treatment as the reviewer suggested, and new data is presented in Figure S5A-B.  As the reviewer suggested to use standard of care drugs, we decided to use cisplatin, as platinum-based drugs are the standard of care for patients with ovarian cancer. As for the metabolic/mTOR drugs that would be tested in combinatorial treatment, we selected representatives of each class:  temsirolimus, CC-223, pevonedistat, AVN944, atorvastatin and pemetrexed. We have chosen two PDCs models for this experiment, one HGSOC (PDC#2) and one LGSOC (PDC#4), as representatives of each ovarian cancer subtype.

First, we investigated the cytotoxic effect of cisplatin alone (Figure S5A) and decided to use a cisplatin concentration of 100uM, which did not show any cytotoxic effect on its own. All other drugs were also used at low concentration that had no significant cytotoxic effect, as to be able to observe the additive cytotoxic effect in the presence of cisplatin. As shown in Figure S5B, pretreatment of cells with cisplatin along with temsirolimus, pevonedistat and AVN944 showed increased cytotoxic effect compared to either single drug treatment alone. We did not observe any enhanced cytotoxicity for drug combinations using CC-223, atorvastatin and pemetrexed (data not shown) in neither of the PDC models.

The results are not surprising to us, but rather aligned with previous findings. Temsirolimus is first generation rapalog that had been clinically tested in ovarian cancer (ref. 79 in the manuscript) and it has shown moderate effect as monotherapy, which is common for all others first generation rapalogs. Therefore, combinatorial treatment should be strongly considered for these first generation rapalogs. The second generation of mTOR inhibitors (AZD8055, vistusertib, dactolisib, sapanisertib, CC-115 and CC-223) displayed already very good cytotoxicity in all PDCs, therefore is very unlikely that we would see enhanced cytotoxicity in combination treatment in our settings. As mentioned above, we did not observe any additive effect between cisplatin and CC-223 (data not shown). Vistusertib, dactolisib and sapanisertib are already used in clinical trials in combinatorial treatments (Ref. 83, 84, 85 in the manuscript), and we show in our study that these metabolic drugs are active as single agents in our preclinical models.

From the metabolic inhibitors, we observed enhanced cytotoxic efficacy in combination with cisplatin for pevonedistat and AVN944.  Both drugs showed moderate (average) activity in several PDCs (Figure 3B), indicative that it would be suitable for combinatorial treatment in our settings and indeed, it worked. Other drugs in our metabolic-target library (erastin, atorvastatin, methotrexate, pemetrexed, TH588, triapine, AZD3965 and disulfiram/CuCl2) showed almost no activity as single treatment in PDCs (Figure 3B), which would suggest that these drugs are not active in ovarian cancer and is likely they will not work in combinatorial treatments. Indeed, combination of cisplatin and atorvastatin or pemetrexed showed no enhance cytotoxicity (data not shown).

Another aspect that we want to underline is that at Helsinki University, we set-up the Precision Medicine Platform (se PMID: 24056683)  with the aim to provide sufficient ex-vivo patient-derived cell culture models (PDCs) that successfully recapitulate the phenotypic signature of the original tumor and use this to assess the drug-response profile of the tumor cells. Our strategy was successful to be implemented for clinical application (see PMID: 33530027). We usually evaluate the drug responses in PDCs as single treatment, but for clinical decisions, usually patients are given one or two chemotherapeutic drugs as standard of care. Is not necessary that we need to evaluate combinatorial treatments in our PDCs, we just identify the most active/potent targeted drugs and the clinicians will design the treatment that normally will include standard of care drugs. 

The reviewer suggested the use of mouse models to validate drug combination. We would like to underline that as described in the discussion part of the manuscript, many of the drugs that we tested in our PDCs models are currently evaluated in clinical trials for ovarian cancer and other solid tumors. Thus, there is no need to restart a preclinical study on mouse models, but rather enlarge the current spectrum of clinical use and this was the aim of our study. Moreover, all the clinical trials where metabolic/mTOR inhibitors were evaluated were focused on HGSOC patients, which is the most prevalent ovarian cancer subtype. In our study, we show that many of these inhibitors work in LGSOC preclinical models as well, which will support future development of clinical trials with LGSOC patients when evaluating these drugs. The strength and novelty of our study is that we established PDCs from 2 different ovarian cancer subtypes, (HGSOC and LGSOC) and we evaluated the potential cytotoxic activity of anti-metabolic and mTOR inhibitors in PDCs as well as in well-established cell lines, to identify subtype-specific drug responses in ovarian cancer.

Given the fact that we managed to establish these PDCs and test their response to a small library of metabolic inhibitors and mTOR targeted drugs and we also provide a comparison with several relevant ovarian cancer cell lines provides a valuable data for researchers interested in ovarian cancer preclinical studies, and will open the field to more studies into ovarian cancer subtypes and drug repurposing.

Reviewer 2 Report

The original article "Evaluating targeted therapies in ovarian cancer metabolism: novel role for PCSK9 and second generation of mTOR inhibi- 3
tors." by Sanz et al. analyzes mTOR inhibitors in the context of the role of PCSK9 in ovarian cancer biology.

Table 1: Please include the references or other information concerning the mutations or cell types.

Figure 1 B and E: Please include +/-SEM and test the significance of the reported differences.

Table 2: Please include the references or other information concerning the used inhibitors.

Figure 2 C and F: Please include +/-SEM.

Author Response

Point by point answer to reviewers:

We would like to thank both reviewers for their insightful comments that strengthened our manuscript and made it more ready for publication. We have addressed all the comments/questions raised by the reviewers and the detailed answers is presented below in red. Also, the new edits made in the manuscript are marked by red for visibility.

Reviewer # 2:

The original article "Evaluating targeted therapies in ovarian cancer metabolism: novel role for PCSK9 and second generation of mTOR inhibitors." by Sanz et al. analyzes mTOR inhibitors in the context of the role of PCSK9 in ovarian cancer biology.

Table 1: Please include the references or other information concerning the mutations or cell types.

Figure 1 B and E: Please include +/-SEM and test the significance of the reported differences.

Table 2: Please include the references or other information concerning the used inhibitors.

Figure 2 C and F: Please include +/-SEM.

Answer to Reviewer #2:

We thank the reviewer for point out these details, which have been addressed entirely.

Table 1: Please include the references or other information concerning the mutations or cell types.

We have included the relevant references and information regarding the mutation types in Table 1.

Figure 1 B and E: Please include +/-SEM and test the significance of the reported differences.

+/-SEM of the reported differences and statistical significance were included in Figures 1B, 1E.

Note that, for statistical significance (t-test), the quantification of WB data is rather difficult to yield a good statistical value, as we have done a relative quantification of each biological replicate as stated in Figure legend. The observed biological effect of higher expression of PCSK9 (cleaved and uncleaved) in HeLa and OVCAR3/OVCAR3cis in Figure 1A/B or in PDC#2, PDC#4 and PDC#5 in Figure 1D/E is evident in each experimental replicates, however, due to WB technical issues (band intensity, antibody staining intensity), the statistical significance was not reached for some samples.

Table 2: Please include the references or other information concerning the used inhibitors.

References for the inhibitors and more information regarding the inhibitor pharmacology have been included in Table 2.

Figure 2 C and F: Please include +/-SEM.

+/-SEM were included in Figures 2C and 2F.

Round 2

Reviewer 1 Report

This is an exciting story with foundational data. However, I still have major concerns as in the first review that pertain to impact of manuscript. The request for mouse studies is not useless since these drugs are in the clinic. Mouse models would allow you to understand and validate the mechanism of action further and effect on omentum vs other sites of tumors. The other way that it could be explored is in co-culture/3 D cultures or organotypic cultures with adipocytes and other cells types in omentum vs peritoneum etc. to begin to understand if drugs validated when part of TME around. Without any new data exploring this mechanism of action, it is difficult to offer any more advice on manuscript.

Author Response

We thank the reviewer for his valuable comments. The academic editor has made his suggestions and we have corrected the manuscript according to his suggestions.